# Efficient Removal of Siloxane from Biogas by Using *β*-Cyclodextrin-Modified Reduced Graphene Oxide Aerogels

**DOI:** 10.3390/nano12152643

**Published:** 2022-07-31

**Authors:** Yanhui Zheng, Xifeng Hou, Siqi Lv, Zichuan Ma, Xiaolong Ma

**Affiliations:** 1Hebei Key Laboratory of Inorganic Nano-Materilas, College of Chemistry and Material Science, Hebei Normal University, Shijiazhuang 050024, China; zhengyh0308@163.com (Y.Z.); xifenghoucc@163.com (X.H.); l15231059127@163.com (S.L.); 2Department of Preshool and Art Education, Shijiazhuang Vocational College of Finance & Economics, Shijiazhuang 050061, China; 3School of Environmental Science and Engineering, Hebei University of Science and Technology, Shijiazhuang 050018, China

**Keywords:** reduced graphene oxide aerogel, *β*-cyclodextrin, hydrothermal reduction, hexamethyldisiloxane, contaminant removal

## Abstract

In this study, *β*-cyclodextrin-modified reduced graphene oxide aerogels (*β*-CD-rGOAs) were synthesized via a one-step hydrothermal method and were used to remove hexamethyldisiloxane (L2) from biogas. The *β*-CD-rGOAs were characterized by the Brunner–Emmet–Teller technique, using Fourier-transform infrared spectroscopy, Raman spectrometry, scanning electron microscopy (SEM), contact angle measurements, and X-ray diffraction. The results of the characterizations indicate that *β*-CD was grafted onto the surface of rGOAs as a cross-linking modifier. The *β*-CD-rGOA had a three-dimensional, cross-linked porous structure. The maximum breakthrough adsorption capacity of L2 on *β*-CD-rGOA at 273 K was 111.8 mg g^−1^. A low inlet concentration and bed temperature facilitated the adsorption of L2. Moreover, the *β*-CD-rGOA was regenerated by annealing at 80 °C, which renders this a promising material for removing L2 from biogas.

## 1. Introduction

Biogas is an increasingly important renewable energy resource that is produced by the digestion of organic materials from sewage treatment plants and waste disposal sites; biogas is comprised primarily of methane (50–70%, *v*/*v*), one of the most widely used energy resources [1,2]. Apart from methane, the other main compound in biogas is CO_2_ (30% to 50%, *v*/*v*), followed by other gases under 2% (e.g., oxygen, nitrogen, hydrogen sulfide, and siloxanes) [3,4]. Among these trace gases, siloxanes have the strongest adverse effect on the utilization of biogas due to the conversion of siloxanes into silicates and microcrystalline quartz during combustion [3], which contributes to abrasion of the inner surfaces of the combustion engine [5,6]. Therefore, the removal of siloxanes must be taken into consideration before biogas applications.

To date, several methods have been developed for removing siloxanes from biogas, including adsorption, absorption, deep chilling, biological treatment, and membrane separation [7,8,9,10]. Among these methods, deep chilling exhibits high removal efficiency and has been applied on an industrial scale; however, this method is too expensive to accept [11]. During absorption, the use of strong acids such as H_2_SO_4_ or HNO_3_ presents many disadvantages, including the corrosion of the equipment, by- and co-product generation, and a negative impact on health and the environment [12]. Regarding biodegradation, the removal efficiency is low, and long processing times are required [13]. Nevertheless, membrane separation is not considered to be a competitive treatment due to the selectivity and durability of the membrane [14]. Adsorption onto porous materials exhibits high removal efficiencies [15,16]. Moreover, it is simple and low-cost [1]. Materials such as activated carbon, silica gel, zeolites, activated alumina, and polymer resins have been used in siloxane purification from biogas [17]. The specific surface and pore structure are key characteristics to obtaining a high siloxane adsorption capacity [18]. In comparison, activated carbons show high removal capacity for siloxanes, but it is important to prolong the lifecycle due to the difficulty of regenerating activated carbon [18,19]. Therefore, there is a need to develop new reusable porous materials that adsorb siloxanes.

Reduced graphene oxide aerogel (rGOA) can be produced by a reduction in graphene oxide (GO), which exhibits higher physicochemical performance compared with GO [20]. rGOA has a three-dimensional (3D) network structure, strong hydrophobicity, and weak polarity because of the low abundance of oxygen-containing functional groups on the graphene sheets [21]. Due to its surface physical properties, rGOAs are mainly used in adsorption, catalysis, energy storage, and other fields [20,22,23]. At present, rGOA is frequently prepared through a direct reduction in GO in a hydrothermal environment. This method does not introduce other reactants to facilitate self-assembly and does not remove the byproducts in the preparation [24]. In addition, rGOA can retain the original physical characteristics of graphene (surface defects and π-conjugated structure) [25]. However, rGOA obtained by direct hydrothermal reduction has low porosity and a small specific surface area. Therefore, it is worthwhile to develop a surface modification to optimize the specific surface area and adsorption capacity.

Cross-linkers, such as polyaniline, polyethylenimine, and *β*-cyclodextrin (*β*-CD), are promising materials for modifying surface chemical properties due to their abundant hydroxyl and amine groups. They can cross-link with rGOA through synergistic interactions of covalent and hydrogen bonding, form a reinforced, 3D porous structure, and increase the specific surface area of rGOA [26,27]. Among these cross-linkers, *β*-CD (as a derivative of cyclodextrin, a biodegradable material produced by hydrolysis of starch by cyclodextrin glucosyltransferase), which is available at a low price and is widely sourced, exhibits extensive degradability and cross-linking [28]. The interior of *β*-CD is hydrophobic, and the surface is hydrophilic (Figure 1). *β*-CD can be grafted onto the surface of rGOAs through interactions with the hydroxyl groups of rGOA by strong hydrogen bonding [29]. Through cross-linking, *β*-CD-modified rGOA (*β*-CD-rGOA) can form a stable 3D structure with many pores, resulting in an enhanced adsorption capacity. Using *β*-CD as a modifier of rGOA results in a high adsorption capacity for naproxen, bisphenol A, and heavy metal ions from an aqueous solution [30,31,32,33]. However, the adsorption of siloxanes has not been reported. Furthermore, most of the rGOAs reported in the literature were prepared with the GO, which is flaky and single-layered, and is synthesized by the Hummer or a modified Hummer method. This procedure increases the complexity of the preparation and the cost. With the wide application of GO, multilayer industrial-grade multilayer graphene (IGGO) has been mass-produced, greatly reducing its cost. However, whether IGGO can be modified by *β*-CD to produce 3D rGOA remains to be studied.

As reported herein, a *β*-CD-rGOA with a 3D structure was synthesized by using IGGO as a raw material via a one-step hydrothermal method. The resultant *β*-CD-rGOA was thoroughly characterized to determine the possible adsorption mechanism. Subsequently, to investigate the optimal adsorption performance of *β*-CD-rGOA for removing hexamethyldisiloxane (L2), batch experiments were designed, including the influence of the *β*-CD dosage, L2 inlet concentration, and bed temperature. Moreover, regeneration experiments were applied to evaluate the recyclability of *β*-CD-rGOA. The results indicate that *β*-CD-rGOA is highly promising in removing siloxanes from biogas.

## 2. Materials and Methods

### 2.1. Chemicals and Reagents

*β*-cyclodextrin (*β*-CD, C_42_H_70_O_35_, Mw 1135 g mol^−^^1^, ρ 1.614 g cm^−^^3^, 99.0% purity, solid powder) was provided by Chengdu Kelon Chemical Reagent Factory (Chengdu, China). Industrial-grade multilayer graphene oxide (IGGO, 95.0% purity, solid powder) was obtained from Suzhou Hengqiu Technology Co., Ltd., (Suzhou, China). Hexamethyldisiloxane (L2, C_6_H_18_OSi_2_, Mw 162 g mol^−1^, ρ 0.764 g cm^−^^3^, 99.0% purity, liquid) was purchased from Aladdin (Shanghai, China). Sairuifu Technology Co., Ltd., (Tianjin, China) supplied absolute ethanol (C_2_H_5_OH, Mw 46 g mol^−^^1^, ρ 0.789 g cm^−^^3^, 99.9% purity, liquid). All reagents were used directly with no further purification. Deionized water was used for all experiments.

### 2.2. Preparation of β-CD-rGOAs

*β*-CD-rGOAs were synthesized via a one-step hydrothermal approach [32]. Briefly, a 60-mL IGGO suspension (4.0 mg mL^−1^) was obtained after ultrasonic treatment for 30 min. Then, the IGGO suspension was added to 120 mg of *β*-CD, with ultrasound for another 30 min. The mixture was enclosed in a 100-mL, Teflon-lined autoclave and heated at 180 °C for 6 h to obtain *β*-CD-reduced graphene-oxide hydrogels (*β*-CD-rGOHs). After washing with deionized water, the hydrogels were dialyzed with 20 *v*/*v*% ethanol for 6 h and freeze-dried for 24 h to obtain rGOA samples with an IGGO/*β*-CD mass ratio of 1:0.5. Various IGGO-to-*β*-CD mass ratios (1:0, 1:0.25, and 1:1) were synthesized by the same method. The products were termed rGOA, *β*-CD-rGOA-0.25, and *β*-CD-rGOA-1, respectively. Some properties of aerogels were shown in Appendix A.

### 2.3. Characterization

The morphology of the as-synthesized materials was characterized via scanning electron microscopy (SEM, S-4800, Hitachi, Tokyo, Japan) and with a Raman spectrometer (Renishaw, NewMills, UK). Transmission electron microscopy (TEM, H-7650, Hitachi, Tokyo, Japan) was used to characterize the fine structure of the materials. The X-ray diffraction (XRD, Bruker AXS, Karlsruhe, Germany) patterns were used to analyze the structural characteristics of the samples with a D8 ADVANCE X-ray source (Bruker AXS, Munich, Germany). The functional group composition was confirmed by Fourier-transform infrared (FTIR, Bruker Optics Co., Karlsruhe, Germany) spectroscopy from 400 to 4000 cm^−1^ with a Tensor 27 spectrometer (Bruker AXS, Karlsruhe, Germany). The N_2_ adsorption–desorption isotherms were measured with an aperture and specific surface area analyzer (Kubo × 1000, Beijing Builder Co., Ltd., Beijing, China) at 77 K and 10^−5^ < *P*/*P*_0_ < 1.0. The Brunner–Emmet–Teller (BET, Beijing Builder Co., Ltd., Beijing, China) specific surface areas (*S*_BET_) were obtained, and the Barrett–Joyner–Halenda method was used to calculate the average pore size (*D*_aver_) and total pore volume (*V*_tot_). The water contact angle was investigated, to determine the hydrophobicity of the samples, with a contact angle/interface system (JY-PHb, Jinhe Instrument Manufacturing Co., Ltd., Nanjing, China). Gas chromatography with a hydrogen flame ionization detector (GC 9790, Fuli Analytical Instrument Co., Ltd., Wenling City, China) was used to observe the L2 concentration with a GDX-102 stationary-phase column (2.0 m long × 2.0 mm inner diameter). The detecting conditions were as follows: oven temperature, 200 °C, injector temperature, 250 °C, and detector temperature, 280 °C.

### 2.4. Adsorption and Regeneration

The L2 dynamic adsorption performance onto the adsorbents was evaluated based on a previous study [34]. The relevant parameters were as follows: L2 inlet concentration (*C*_in_), 14.62 mg L^−1^, gas flow rate (*V*_g_), 50 mL min^−1^ (superficial velocity of 100 cm min^−^^1^, cross-sectional area of 0.5 cm^2^), and temperature (*T*), 20 °C. When the L2 outlet concentration (*C*_out,t_) equaled *C*_in_, the adsorption reached saturation. The association between *C*_out,t_/*C*_in_ and time (*t*) was used to express breakthrough curves that were used to describe the dynamic adsorption performance. Usually, the adsorption behavior was evaluated by the following two quantities: breakthrough time (*t*_B_, min) and breakthrough adsorption quantity (*Q*_B_, mg g^−1^). The breakthrough time was defined as the time in which *C*_out,t_/*C*_in_ = 0.05, and *Q*_B_ as the adsorption quantity was when the adsorption time was *t*_B_.

The adsorption capacity (*Q*_t_), corresponding to *t*, was evaluated with Equation (1) [35]:(1)$$Q_{t}=\frac{V_{g}C_{\mathrm{in}}}{m}\int_{0}^{t}\left(1-\frac{C_{\mathrm{out},t}}{C_{\mathrm{in}}}\right)tt,$$
where *C*_in_ and *C*_out,t_ are the inlet concentration (mg L^−1^) and outlet concentration (mg L^−1^) at adsorption time *t* (min), *m* (g) is the mass of adsorbent, and *V*_g_ (L min^−1^) is the gas flow rate. By substituting *C*_out,t_/*C*_in_ = 0.05 in Equation (1), *Q*_B_ is obtained.

When the *β*-CD-rGOAs were saturated with L2, the regeneration of the used *β*-CD-rGOAs was performed in a water bath at 80 °C under blowing nitrogen for 30 min. This adsorption–desorption was repeated 10× to evaluate the regenerative performance of the *β*-CD-rGOAs. The continuous adsorption–desorption cycle was repeated for the first 5 times, and an adsorption–desorption cycle was repeated daily for the next 5 times.

### 2.5. Mathematical Models

The Yoon–Nelson model was selected to predict the theoretical breakthrough adsorption capacity (*Q*_B,th_, mg g^−1^) and the theoretical breakthrough time (*t*_B,th_, min) and to better understand adsorption phenomena of L2 in the *β*-CD-rGOAs. The model is represented by Equation (2) [36]:(2)$$\frac{C_{\mathrm{out},t}}{C_{\mathrm{in}}}=\frac{1}{1+\exp[K_{\mathrm{YN}}(\tau -t)]}\times 100\%$$
where *K*_YN_ is the Yoon–Nelson constant of the model, *t* is the adsorption time (min), and *τ* is the time when the ratio *C*_out,t_/*C*_in_ = 0.5.

The model values (*K*_YN_ and *τ*) of various adsorption experiments were obtained through simulation using Equation (2). The *t*_B,th_ was obtained by the Equation (2) when the *C*_out,t_/*C*_in_ ratio was 0.05.

Hence, substituting Equation (2) in Equation (1), the *Q*_B,th_ at *t*_B,th_ can be calculated.

## 3. Results

### 3.1. Effect of Modifier β-CD

Appendix A shows digital photos of *β*-CD-rGOHs and *β*-CD-rGOAs. After hydrothermal reduction, the hydrogels were prepared, indicating that reduction-induced self-assembly facilitated the formation of hydrogels. Figure 2 shows the nitrogen adsorption/desorption isotherms and the pore size distributions of IGGO, rGOA, and *β*-CD-rGOA-0.5. In accordance with the IUPAC classification, *β*-CD-rGOA-0.5 demonstrated the features of type IV isotherms with H3 hysteresis loops, which is indicative of a mesoporous structure [37]. Alternatively, IGGO and rGOA followed a type III isotherm with a macroporous or nonporous material and low BET surface area. Table 1 shows the textural parameters of IGGO and *β*-CD-rGOAs. The BET analysis indicates that the specific surface areas of IGGO, rGOA, *β*-CD-rGOA-0.25, *β*-CD-rGOA-0.5, and *β*-CD-rGOA-1 were 7.4, 55.2, 59.3, 163.5, and 112.3 m^2^ g^−1^, respectively. The increased BET area of *β*-CD-rGOAs was possible due to the reduction-induced self-assembly and *β*-CD crosslinking effect. With increasing *β*-CD loading, the BET area increased, and the porous structure was enhanced. The pore size distributions indicate that the primary pores of *β*-CD-rGOA-0.5 were within a diameter of 2–6 nm, which mainly contributed to the specific surface area. However, more *β*-CD caused a decrease in the surface area and total pore volume, which might be attributable to the excessive aggregation of *β*-CD [32]. Therefore, *β*-CD-rGOA-0.5 had an appropriate amount of *β*-CD. Based on the literature, a narrower mesoporous range facilitates L2 adsorption [38].

Appendix A shows a water droplet on the surface of a *β*-CD-rGOA-0.5 film, which indicates the strong hydrophobicity of the material. Table 1 shows the experimental values of the contact angle (θ) for the studied materials. The *β*-CD-rGOAs all exhibited high hydrophobicity, which is attributable to the removal of some oxygen-containing functional groups because of the hydrothermal reaction.

### 3.2. Characterization of Adsorbents

Figure 3 shows SEM and TEM images of IGGO, rGOA, and *β*-CD-rGOA-0.5. IGGO had a relatively smooth surface with no wrinkles and holes. After hydrothermal reduction, the folds of rGOA and *β*-CD-rGOA-0.5 increased in number, and the surfaces were not smooth. Thinner lamellae and more pores can be identified in the SEM and TEM images, and are shown in Figure 3c–f. In contrast, *β*-CD-rGOA-0.5 exhibited more wrinkles than rGOA, implying that it may possess a higher specific surface area and larger pore volume, which could be due to the modification of the *β*-CD crosslinking effect. These folds could improve the surface area and increase the adsorption [39]. These results are consistent with the previous findings shown in Table 1.

Figure 4 shows XRD diagrams of IGGO, rGOA, and *β*-CD-rGOA-0.5. An extensive alteration of the structures was noted by the XRD patterns. A sharp diffraction peak at 11.6° in the IGGO pattern corresponded to the (001) plane, which reveals a highly ordered structure with interlayer spacing of about 7.6 Å [40]. Moreover, a narrow peak (100) centered at 42.7° was observed for IGGO, corresponding to the long range order in the graphitic planes. After hydrothermal reduction, the characteristic peak of IGGO disappeared, and characteristic diffraction peaks indexed to (002) facets of graphitic carbon for rGOA and *β*-CD-rGOA-0.5 at 24.9° and 23.4°, respectively; these corresponded to interlayer spacings of 3.7 and 3.9 Å, respectively [38]. The presence of the broad diffraction peak signified the disordered structure, and the loss of 42.7° could reflect the loss of planarity for rGOA and *β*-CD-rGOA-0.5. The interlayer spacing of the rGOA and *β*-CD-rGOA-0.5 were smaller than that of IGGO due to the removal of oxygen-containing functional groups and π–π interactions during self-assembly [38]. The interlayer spacing of *β*-CD-rGOA-0.5 was slightly larger than that of rGOA, which is attributable to the cross-linking modification of *β*-CD. When *β*-CD was introduced into the *β*-CD-rGOAs, the crosslinking effect of more hydrogen bonds suppressed π–π restacking, which enlarged the interlamellar spacing.

Raman spectroscopy is frequently used to characterize the deficiencies in graphene-based materials (Figure 5). The Raman spectra indicate two distinct characteristic peaks in IGGO, rGOA, and *β*-CD-rGOA-0.5 at 1600 and 1345 cm^−1^, corresponding to the *G* and *D* peaks, respectively, of carbon materials. The *G* peak is the characteristic feature for graphite (*sp*^2^-hybridized carbon in-plane stretching vibration), and the *D* peak is due to the disordered carbon, including, e.g., *sp*^3^-hybridized carbon and dopant atoms. Usually, the D/G band intensity ratio (*I*_D_/*I*_G_) represents the degree of disorder in carbon [41]. The *I*_D_/*I*_G_ of rGOA (0.95) and *β*-CD-rGOA-0.5 (0.98) indicate a noticeable increase compared with IGGO (0.83). This indicates that the hydrothermal reduction increased the number of defects and resulted in a low degree of carbonization. *β*-CD-rGOA-0.5 had a slightly higher value of *I*_D_/*I*_G_ than rGOA, which indicates that introducing *β*-CD lead to an increase in the degree of defects. Because of the hydrogen bonding between *β*-CD and rGOA, the degree of graphitization was decreased, which led to a higher *I*_D_/*I*_G_ ratio. The peaks located in the range of 2300–2800 cm^−1^ are the 2D bands, which are another characteristic peak of graphene [37]. In comparison with IGGO, rGOA and *β*-CD-rGOA-0.5 exhibited a fairly broad and up-shift 2D peak in the Raman spectrum, demonstrating its low-layer structure, which was consistent with the TEM results.

Figure 6 shows the FTIR spectroscopy results for IGGO, rGOA, and *β*-CD-rGOA-0.5. The broad peaks at 3441 cm^−^^1^ are attributable to the O–H stretching vibration in the carboxyl groups, C–OH groups, and adsorbed-state water. In the IGGO spectrum, 2927 cm^−1^ (low-intensity stretching and bending vibrations from C–H), 1727 cm^−1^ (C=O stretching vibration in the carboxyl groups), 1600 cm^−1^ (C=C vibration of the graphene skeleton), and 1050 cm^−1^ (C–O–C stretching vibration in the epoxy groups) were observed, which indicates that IGGO has a large number of oxygen-containing groups on its surface [36]. After hydrothermal reduction, the C–O–C band at 1050 cm^−1^ and the C=O peak at 1727 cm^−1^ in the rGOA were no longer evident, which indicates that some of the oxygen-containing groups in IGGO underwent reduction. However, after adding *β*-CD, the FTIR spectra of *β*-CD- rGOA-0.5 indicate that, not only was the C=O peak absent, but the spectral features of *β*-CD were evident (the coupled C–O–C stretching/O–H bending vibrations at 1163 cm^−1^ and the coupled C–O/C–C stretching/O–H bending vibrations at 1050 cm^−1^ and 1112 cm^−1^), indicating the modification of rGOA with *β*-CD [41,42]. Moreover, the broader absorption of *β*-CD-rGOA-0.5 over the range of 3000–3750 cm^−1^ (high wavenumbers) was evident, which is indicative of hydrogen bond formation between rGOA and *β*-CD [43].

In accordance with the experimental results, a possible working mechanism was elucidated (Figure 7). In a 180 °C hydrothermal enviroment, carbonyl groups can be efficiently reduced to hydroxyl groups, and the ring-opening reaction of an epoxy group occured to form a hydroxyl group. After the modification of *β*-CD on the rGOA surface, the π–π stacking between the graphene sheets can be weakened; with the hydroxyl group generated, the H-bonds would be increased. When hydroxyl segments at the edge and plane of rGOA interact through hydrogen bonding, a porous 3D structure is possibly facilitated. In this manner, *β*-CD-rGOA-0.5 can provide more active sites and channels, whereas the adsorption ability was increased.

### 3.3. Comparison of Dynamic Adsorption Performances

The adsorption breakthrough curves by using the Yoon–Nelson model of the IGGO, rGOA, *β*-CD-rGOA-0.25, *β*-CD-rGOA-0.5, and *β*-CD-rGOA-1 (Figure 8) were tested at 20 °C, *V*_g_ of 50 mL min^−1^, and *C*_in_ of 14.62 mg L^−1^ for comparing the dynamic adsorption performance. Table 2 shows the experimental adsorption values and the calculated model parameters. The high correlation coefficients (*R*^2^ > 0.99) indicate that the Yoon–Nelson model fit well with the L2 dynamic adsorption onto *β*-CD-rGOAs. Thus, in the following discussion, the calculated theoretical parameter values (*t*_B,th_ and *Q*_B,th_) were used to compare and analyze the adsorption capacity of the *β*-CD-rGOAs. The L2 directly penetrated the IGGO, indicating that the IGGO had virtually no adsorption ability. Furthermore, the adsorption performance among the *β*-CD-rGOAs was ranked as *β*-CD-rGOA-0.5 > *β*-CD-rGOA-1 > *β*-CD-rGOA-0.25 > rGOA, and the *β*-CD-rGOA-0.5 exhibited the maximum breakthrough time and adsorption capacity, with *t*_B,th_ and *Q*_B,th_ values of 11.03 min and 88.7 mg g^−1^, respectively. To determine the factors that are responsible for the *β*-CD-rGOA adsorption capacity, the relationships between *Q*_B,th_ and *S*_BET_, *V*_tot_, and the contact angle were evaluated (Figure 9). The linear regression equation was y = a + bx. Some important statistical data were shown in Appendix A. Specifically, the closer *R^2^* is to 1, the better the linear relationship; the smaller the standard deviation, the more precise the data. Accordingly, the correlation coefficient (*R*^2^) values for *Q*_B,th_–*S*_BET_, *Q*_B,th_–*V*_tot_, and *Q*_B,th_–contact angle were 0.9953, 0.9507, and 0.8563, respectively. Therefore, it can be broadly concluded that *S*_BET_ and *V*_tot_ were the predominant factors that affected the L2 adsorption capacity, while contact angle also has a certain degree of positive correlation [38].

### 3.4. Influence of Inlet Concentration and Bed Temperature

Inlet concentration is a significant factor influencing the adsorption process [44]. Figure 10 shows the curve of the *t*_B,th_ and *Q*_B,th_ change over *C*_in_. The *t*_B,th_ decreased exponentially with *C*_in_. However, *Q*_B,th_ values presented, at first, an increasing trend and then a decreasing one in the concentration series experiment. As reported [45,46], it might be related to the co-action of the adsorption driving force of L2 molecules (positive correlation to *C*_in_) and the accessibility of adsorption sites (negative correlation to *C*_in_), which could portend L2 adsorption behavior during industrial application.

Bed temperature is another important factor that affects the L2 adsorption efficiency. Breakthrough curves fitted by the Yoon–Nelson model of L2 adsorption on *β*-CD-rGOA-0.5 at different bed temperatures (0–55 °C) were shown in Figure 11, and the details of the calculated results by the Yoon–Nelson model were presented in Table 3. There was a tendency that, the higher the bed temperature, the shorter the *t*_B,th_, indicating that low temperature was beneficial to the adsorption of L2 on *β*-CD-rGOA-0.5. The results revealed that the L2 adsorption on *β*-CD-rGOA-0.5 was exothermic. The maximum breakthrough adsorption capacity (111.8 mg g^−1^) of L2 on *β*-CD-rGOA-0.5 was observed at a temperature of 0 °C, *C*_in_ of 14.62 mg L^−1^, and *V*_g_ of 50 mL min^−1^.

### 3.5. Recycling Performance of β-CD-rGOA-0.5

The recycling performance is a key factor for evaluating practical applications of an adsorbent [47]. Thus, the spent *β*-CD-rGOA-0.5 was assessed via thermal treatment at 80 °C for 30 min, and, subsequently, through five adsorption–desorption cycles. Figure 12 shows the adsorption breakthrough curves for L2 of *β*-CD-rGOA-0.5 after each desorption, and Appendix A shows that there is no residual L2 on the *β*-CD-rGOA-0.5 after continuous cycles. These results indicate that *β*-CD-rGOA-0.5 can be completely reused or recycled. Therefore, the as-prepared *β*-CD-rGOA-0.5 is a promising candidate for a number of applications on an industrial scale.

## 4. Conclusions

*β*-CD-rGOA-0.5, an efficient adsorbent material, was synthesized by a one-step hydrothermal reaction with *β*-CD as a surface modification agent. The *β*-CD-rGOA-0.5 exhibited a stable, hierarchically porous, 3D structure because of the many hydrogen bonds and the crosslinking effect. Furthermore, *β*-CD-rGOA-0.5 exhibited many useful properties, including a high surface area (163.5 m^2^ g^−1^), total pore volume (0.68 cm^3^ g^−1^), and water contact angle (128.4°). Moreover, the maximum breakthrough adsorption capacity of L2 onto *β*-CD-rGOA-0.5 was 111.8 mg g^−1^ at a temperature of 0 °C, *C*_in_ of 14.62 mg L^−1^, and *V*_g_ of 50 mL min^−1^. In addition, the *Q*_B,th_ was strongly linearly proportional to *S*_BET_, *V*_tot_, and the contact angle, which indicates that the adsorption mechanism of *β*-CD-rGOA-0.5 for L2 might depend on capillary condensation and the hydrophobic effect. Furthermore, the low bed temperature and inlet concentration resulted in high L2 adsorption levels for *β*-CD-rGOA-0.5. Finally, the *β*-CD-rGOA-0.5 can be easily regenerated and reused multiple times by heating at 80 °C for 30 min. All of these characteristics indicate that the *β*-CD-rGOA-0.5 exhibits prospects of industrial application. The present study provided experimental evidence and a theoretical basis for the removal of siloxane from actual biogas. Next, we will investigate the effects of coexisting substances in biogas on siloxane adsorption. Further studies directly addressing practical problems are needed.

## Figures and Tables

**Figure 1 nanomaterials-12-02643-f001:**
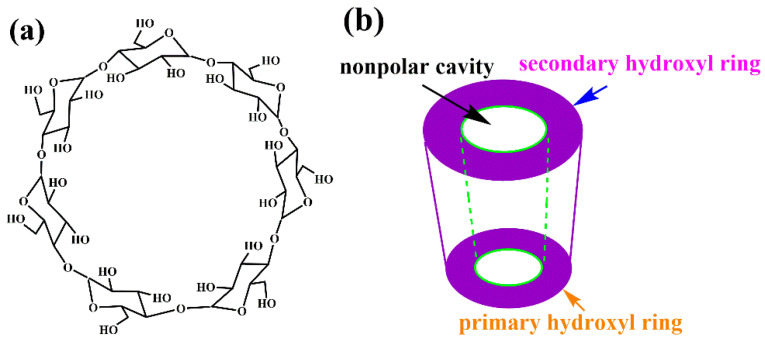
Molecular structure (**a**) and molecular shape (**b**) of *β*-CD.

**Figure 2 nanomaterials-12-02643-f002:**
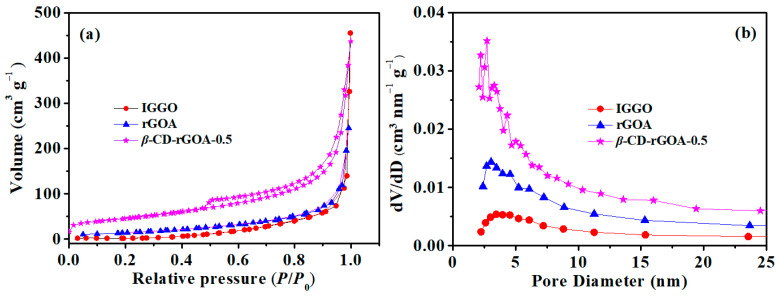
Comparison of the adsorption–desorption isotherms of nitrogen (**a**) and pore size distribution (**b**) by IGGO, rGOA, and *β*-CD-rGOA-0.5.

**Figure 3 nanomaterials-12-02643-f003:**
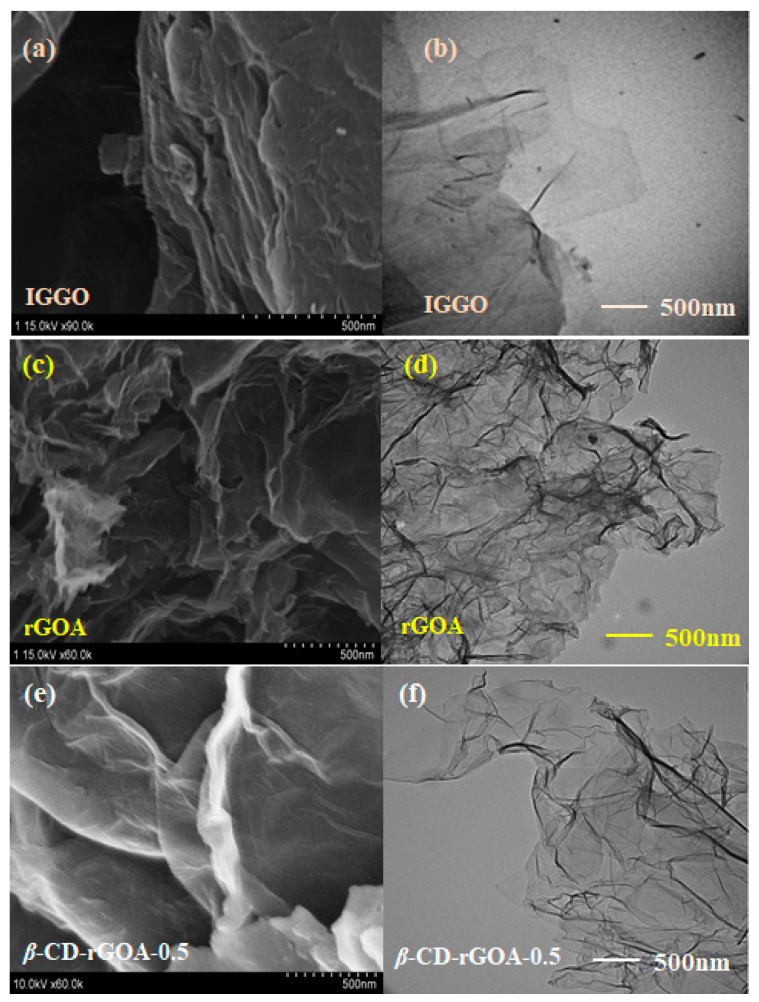
SEM images (**a**,**c**,**e**) and TEM images (**b**,**d**,**f**) of IGGO, rGOA, and *β*-CD-rGOA-0.5.

**Figure 4 nanomaterials-12-02643-f004:**
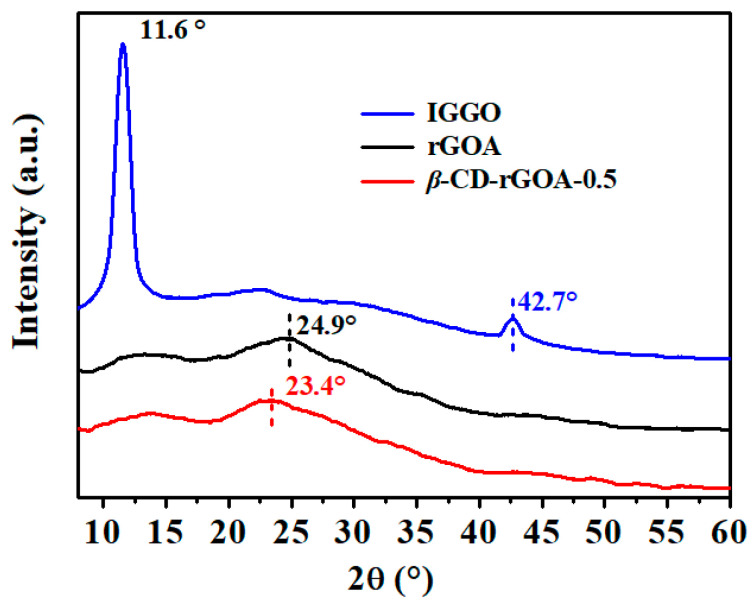
XRD patterns of IGGO, rGOA, and *β*-CD-rGOA-0.5.

**Figure 5 nanomaterials-12-02643-f005:**
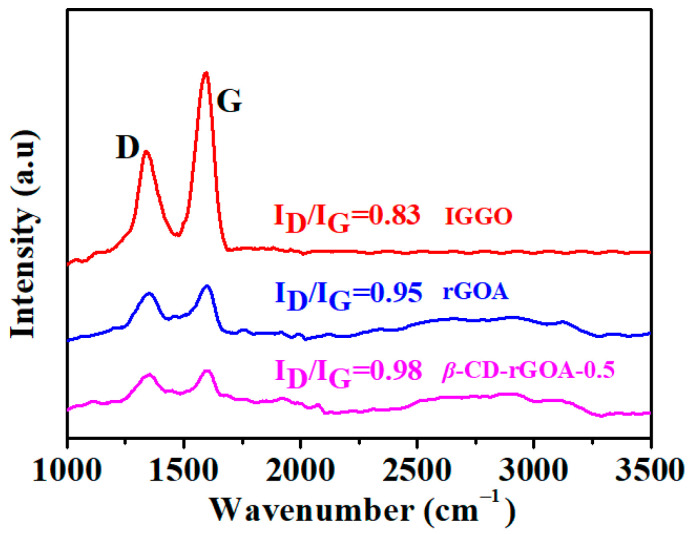
Raman spectra of IGGO, rGOA, and *β*-CD-rGOA-0.5.

**Figure 6 nanomaterials-12-02643-f006:**
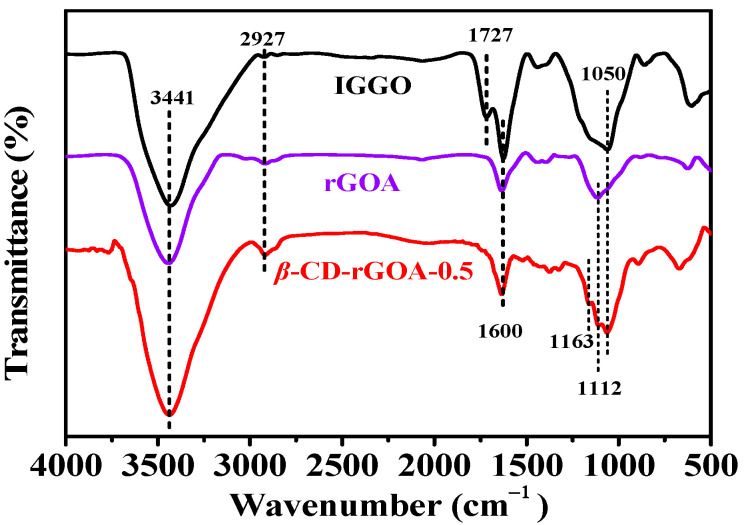
FTIR spectra of IGGO, rGOA, and *β*-CD-rGOA-0.5.

**Figure 7 nanomaterials-12-02643-f007:**
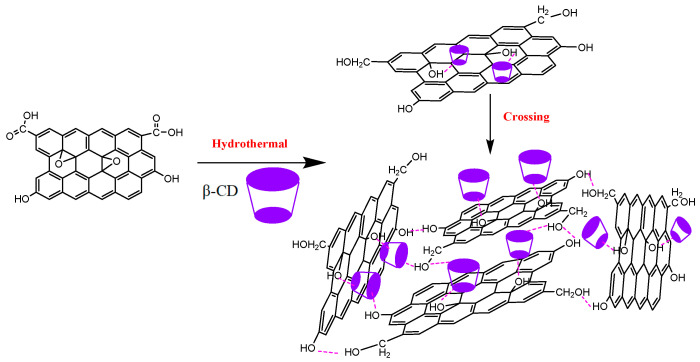
Working mechanism of hydrothermal reduction and crosslinking.

**Figure 8 nanomaterials-12-02643-f008:**
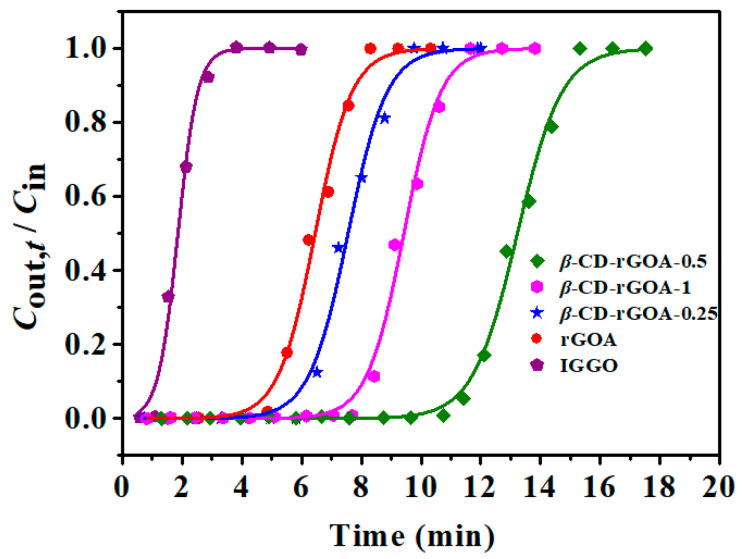
Breakthrough curves of different adsorbents for L2.

**Figure 9 nanomaterials-12-02643-f009:**
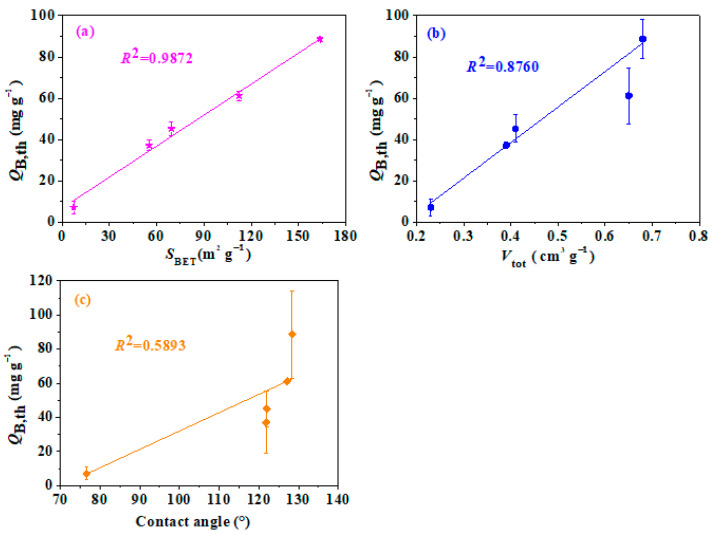
Relationship between *Q*_B_ and *S*_BET_ (**a**), *Q*_B_ and *V*_tot_ (**b**), *Q*_B_ and contact angle (**c**) for *β*-CD-rGOAs adsorbents.

**Figure 10 nanomaterials-12-02643-f010:**
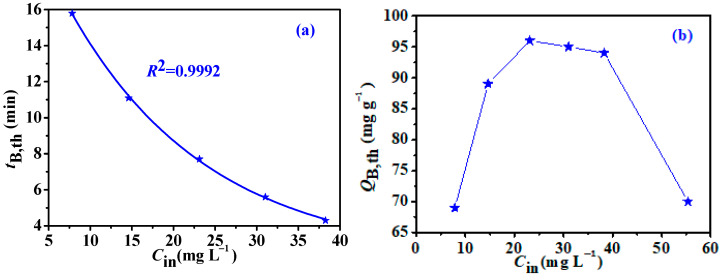
Effect of inlet concentration on *t*_B,th_ (**a**) and *Q*_B,th_ (**b**).

**Figure 11 nanomaterials-12-02643-f011:**
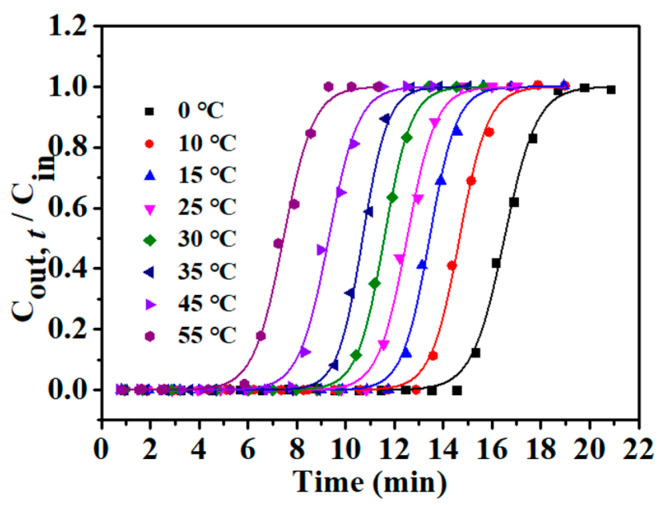
Adsorption breakthrough curves for *β*-CD-rGOA-0.5 at different temperatures by Yoon–Nelson model.

**Figure 12 nanomaterials-12-02643-f012:**
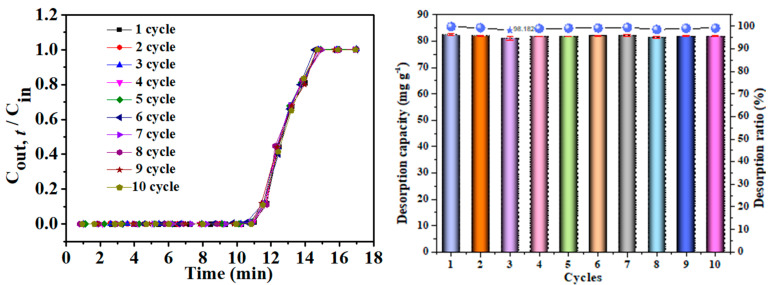
Adsorption breakthrough curves of L2 and *β*-CD-rGOA-0.5 desorption rate after cycles.

**Table 1 nanomaterials-12-02643-t001:** Properties of IGGO and *β*-CD-rGOAs.

Sample	Specific Surface Area (m^2^ g^−^^1^)	Total Pore Volume (cm^3^ g^−1^)	Average Pore Diameter (nm)	Contact Angle (°)
IGGO	7.4	0.23	13.18	76.6
rGOA	55.2	0.39	9.04	121.8
*β*-CD-rGOA-0.25	69.3	0.41	7.28	122.0
*β*-CD-rGOA-0.5	163.5	0.68	5.93	128.4
*β*-CD-rGOA-1	112.3	0.65	7.87	127.1

**Table 2 nanomaterials-12-02643-t002:** Adsorption parameters of different adsorbents for L2 ^a^.

Absorbent	*t*_B_ (min)	*t*_B,th_ (min)	*Q*_B,th_ (mg g^−^^1^)	*K_YN_*	*τ* (min)	*R^2^*
IGGO	1.16	0.89	7.1	3.0263	1.86	0.9901
rGOA	5.14	4.64	37.1	1.6239	6.45	0.9941
*β*-CD-rGOA-0.25	5.78	5.64	45.2	1.5359	7.55	0.9934
*β*-CD-rGOA-0.5	11.27	11.03	88.7	1.3434	13.22	0.9952
*β*-CD-rGOA-1	8.00	7.62	61.2	1.6331	9.42	0.9938

^a^*m* ≈ 0.10 g, *C*_in_ = 14.62 mg L^−1^, *V*_g_ = 50 mL min^−1^.

**Table 3 nanomaterials-12-02643-t003:** Influence of the temperature on the adsorption of L2 over *β*-CD-rGOA-0.5 ^a^.

Temp. °C	*t*_B,th_ min	*Q*_B,th_ mg g^−1^	*K_YN_*	*τ* min	*R^2^*
0	14.56	111.8	1.5189	16.50	0.9977
10	12.94	96.3	1.7121	14.66	0.9978
15	11.80	91.5	1.8068	13.43	0.9975
25	10.78	82.7	1.7031	12.51	0.9963
30	9.95	75.5	1.8085	11.57	0.9989
35	9.18	74.4	1.9482	10.69	0.9974
45	7.59	61.54	1.6291	9.38	0.9941
55	5.60	45.96	1.5249	7.46	0.9938

^a^*m* ≈ 0.10 g, *C*_in_ = 14.62 mg L^−1^, *V*_g_ = 50 mL min^−1^.

## Data Availability

Data is contained within the article or Appendix A. The data presented in this study are available in Appendix A.

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
