# Peer review of "Efficient Removal of Siloxane from Biogas by Using β-Cyclodextrin-Modified Reduced Graphene Oxide Aerogels"

_nanomaterials, 2022, doi:10.3390/nano12152643_

Round 1

Reviewer 1 Report

This manuscript entitled "Efficient Removal of Siloxane from Biogas by Using β-Cyclodextrin-Modified Reduced Graphene Oxide Aerogels'' demonstrates the preparation of β-cyclodextrin-modified reduced graphene oxide aerogels aiming to remove hexamethyldisiloxane from biogas, through adsorption.

Although the concept is interesting and novel, the manuscript is not well-written and contains too many flaws, while its English needs to be improved.

I recommend the rejection of the manuscript, but I encourage the authors to address my concerns and to resubmit it.

My comments:

1)   The samples name is a confusion! Why β-CD-rGOA-0 is not simply named as rGOA? What does A represent? It needs to be addressed.

2)   In Figure 2a, the symbols confuse the reader, too. What do the empty stars, circles and triangles represent?

3)   In line 217, XRD is not a spectrum! Please correct.

4)   In Figure 4, the provided XRD patterns should be provided until 60o. There is another small peak at ~45o that corresponds to rGO like structures which is not visible in this case. In addition, the authors have to comment on the peaks’ width at 24.9°, and 23.4°. How did the authors determine the interlayer distance of the RGO composites?

5)   In Figure 5, the key 2D peak at ~2500 cm-1 is missing. Raman spectra should be reconducted.

6)   In Figure 6, since the peak at ~3400 cm-1 that corresponds to -OH remains too intense (in particular it increases due to the presence of -OH by β-CD), I do not agree that reduction is obvious. In this context, I consider XPS mandatorily should be provided.

7)   Characterization for the composite β-CD-rGOA-0.25 is incomplete (Raman, XRD, FT-IR are missing) compared to the rest of composites.

8)   Thermal properties of the composite aerogels should be provided.

9)   Indicative characterization of the composites upon the adsorption of L2 should be provided.

10) I strongly disagree with the provided surface properties of the composite β-CD-rGOA-0.5. The authors have to explain about its strong hydrophobicity. Is it normal? The surface of Β-CD contains too many hydroxyl groups!  

Reviewer 2 Report

This manuscript present experimental results on the synthesis and testing of a novel sorbent for removal of undesirable gases from biogas. The other is well reported, and the article is easy to read and well organized. Some revisions are needed to improve quality and depth of presentation, and to improve the conclusions and outlook of the study, as follows:

1) Section 2.1: to aid new researchers to this reaction, it would be useful to provide information about the physical state of the reagents, and any other relevant property besides purity, such as density, molecular weight, etc.

2) Line 120: ADVANCE.

3) Section 2.4: Not clear what the "gas" was that L2 was mixed with. Was it biogas? Did it at least have any biogas component? It would be valuable to know how selective the sorbent is for L2, or at least if any other component of biogas can interfere with the adsorption.

4) Line 138: a volumetric flow rate does not have much physical meaning without cross-sectional area, this could be very fast or very slow. Superficial velocity, Reynold's number, or some other relevant flow parameter for packed columns or porous media might be more useful.

5) Line 153: is 5 times enough regenerations? In practice, these materials would be regenerated how often? Daily? Weekly? Definitely a lot more than 5 times.

6) Figure 3: images like this are quite inconclusive, and only a few images show what the author wants to show... Would TEM provide any other relevant detail that SEM cannot see?

7) Figures 9a and 9b: what are the origins of these fitted lines. Should they go through (0,0)? That is, if no surface are and no pore volume, no adsorption capacity? If this is not what was done, justification is needed. In any case, it is not clear what fitting rule was used other than maximizing R-squared.

8) Figure 9c: here, the origin choice is more difficult to make, would it be (90,0)? Also, should this be fitted with a straight line? The data points may suggest an exponential relationship. More mechanistic consideration is needed for this analysis.

9) Section 3.4 and Figure 10: it is obvious that when C,in increases, the breakthrough time would decrease. This does not mean that low concentration is better. The more correct measurement should be the rate of adsorption. Higher C,in likely accelerates adsorption, but this is not easily evident from breakthrough studies. A more rigorous mass transfer-based or kinetics-based approach is needed to determine how concentration affects the adsorption process. Basically, it is desirable to know what incremental volume of sorbent is needed to remove an incremental amount of L2 from the biogas.

10) Figure 11: are these temperatures relevant for the intended application? That is, is biogas relatively "cool"? Why not test higher temperatures that may be seen in the typical biogas generation conditions?

11) Section 3.5: as mentioned before, 5 cycles is quite a short study on regeneration. Also, material characterization after the last cycle would be useful to point to what is making performance degrade: are pores closing, is the material degrading chemically, is the XRD structure being lost, etc. Certainly SEM would not tell much...

12) Section 4: this only summarizes the work and results. There should be actual conclusions in terms of the next research steps needed, the potential for industrial use, the need to test under more real conditions, etc.

Reviewer 3 Report

The work is not innovative, porous materials have been investigated for decades, I recommend a more in-depth study of the literature that does not stop at 10 years ago. However, the solution seems to have a good response from the point of view of efficiency. A 3d material is proposed, it would be appropriate to show macorscopic images of the solution obtained.

Reviewer 4 Report

In this work, an efficient removal of siloxane from biogas by using β-cyclodextrin-modified reduced graphene oxide aerogels. The idea of this research is interesting to readers. The background is well studied and the presentation of the method is very clear and sound, but there are some minor issues to be addressed:

The author should study the TEM analysis of IGGO, β-CD-rGOA-0, and β-CD-rGOA-0.5

The author should discuss elaborately in the working mechanism of reduction and crosslinking of β-CD on the rGOA surface.

In section 2.2. Preparation of β-CD-rGOAs, the suitable reference should be cited.

The author should provide more discussion in the Figure 3, SEM images of IGGO, β-CD-rGOA-0, and β-CD-rGOA-0.5.

Round 2

Reviewer 1 Report

Although XPS analysis would further enhance the reliability of the manuscript, I recommend its publication in the current form, since the authors have adequately addressed my comments.

Reviewer 2 Report

The authors have satisfactorily addressed my suggestions and questions, and the revisions made have improved the quality and depth of the study.

Reviewer 3 Report

thanks for the explanations